# Ultrasound-Guided Femoral Nerve Block in Geriatric Patients with Hip Fracture in the Emergency Department

**DOI:** 10.3390/jcm11102778

**Published:** 2022-05-14

**Authors:** Tou-Yuan Tsai, Kar Mun Cheong, Yung-Cheng Su, Ming-Chieh Shih, Su Weng Chau, Mei-Wen Chen, Chien-Ting Chen, Yi-Kung Lee, Jen-Tang Sun, Kuan-Fu Chen, Kuo-Chih Chen, Eric H. Chou

**Affiliations:** 1Emergency Department, Dalin Tzu Chi Hospital, Buddhist Tzu Chi Medical Foundation, Chiayi 62247, Taiwan; 96311123@gms.tcu.edu.tw (T.-Y.T.); athrun_sasuke@hotmail.com (K.M.C.); drsu119@gmail.com (Y.-C.S.); suweng82@gmail.com (S.W.C.); lyg1968@gmail.com (Y.-K.L.); 2School of Medicine, Tzu Chi University, Hualien 97004, Taiwan; tangtang05231980@hotmail.com; 3Institute for Medical Engineering and Science, MIT, Cambridge, MA 02142, USA; littlecanargie@gmail.com; 4Department of Nursing, Dalin Tzu Chi Hospital, Buddhist Tzu Chi Medical Foundation, Dalin, Chiayi 62224, Taiwan; kay3913@gmail.com (M.-W.C.); smile101164@gmail.com (C.-T.C.); 5Department of Emergency Medicine, Far Eastern Memorial Hospital, New Taipei City 22060, Taiwan; 6Department of Emergency Medicine, Chang Gung Memorial Hospital, Keelung 20401, Taiwan; drkfchen@gmail.com; 7Clinical Informatics and Medical Statistics Research Center, Chang Gung University, Taoyuan 33323, Taiwan; 8Community Medicine Research Center, Chang Gung Memorial Hospital, Keelung 20401, Taiwan; 9Department of Emergency Medicine, Shuang Ho Hospital, Taipei Medical University, New Taipei City 23561, Taiwan; juice119@gmail.com; 10Department of Emergency Medicine, Baylor Scott & White All Saints Medical Center, Fort Worth, TX 76104, USA; 11Department of Emergency Medicine, Baylor University Medical Center, Dallas, TX 76104, USA

**Keywords:** opioid, pain, regional anesthesia, ultrasound, emergency department

## Abstract

Background and Objectives: Systemic analgesics, including opioids, are commonly used for acute pain control in traumatic hip fracture patients in the emergency department (ED). However, their use is associated with high rates of adverse reactions in the geriatric population. As such, the aim of this study was to investigate the impact of lidocaine-based single-shot ultrasound-guided femoral nerve block (USFNB) on the standard care for acute pain management in geriatric patients with traumatic hip fracture in the ED. Methods: This retrospective, single-center, observational study included adult patients aged ≥60 years presenting with acute traumatic hip fracture in the ED between 1 January 2017 and 31 December 2020. The primary outcome measure was the difference in the amount of opioid use, in terms of morphine milligram equivalents (MME), between lidocaine-based single-shot USFNB and standard care groups. The obtained data were evaluated through a time-to-event analysis (time to meaningful pain relief), a time course analysis, and a multivariable analysis. Results: Overall, 607 adult patients (USFNB group, 66; standard care group, 541) were included in the study. The patients in the USFNB group required 80% less MME than those in the standard care group (0.52 ± 1.47 vs. 2.57 ± 2.53, *p* < 0.001). The multivariable Cox proportional hazards regression models showed that patients who received USFNB achieved meaningful pain relief 2.37-fold faster (hazard ratio (HR) = 2.37, 95% confidence intervals (CI) = 1.73–3.24, *p* < 0.001). Conclusions: In geriatric patients with hip fractures, a lidocaine-based single-shot USFNB can significantly reduce opioid consumption and provide more rapid and effective pain reduction.

## 1. Introduction

Hip fracture is one of the most common causes of emergent anesthesia and surgery in the elderly [1]. In a nationwide study, one third of the elderly were reported to have experienced at least one fall in a year, some of which caused hip fractures [2,3]. It is also well known that hip fracture is associated with an increased risk of major morbidity and mortality [4]; the one-year mortality rate after hip fracture is reported to be 22% [5].

Acute pain management in the geriatric population with hip fracture is challenging, and contraindications and comorbidities may lead to delayed recovery [6,7]. The use of nonsteroidal inflammatory drugs for pain management can cause kidney injury and gastrointestinal bleeding, and these drugs are associated with high rates of hypersensitivity reactions [8]. Opioids are generally considered to be a standard pharmacologic intervention for pain management in patients with traumatic hip fracture [9]. However, the known adverse effects of opioids include nausea, vomiting, sedation, cognitive impairment, and respiratory depression, which frequently occur in the elderly population. Opioids have a high potential for being abused and misused, which, along with failed health and commercial policies, has caused epidemics in several countries. A previous study has shown that the prevalence of preoperative opioid use in the geriatric population is high [10]. The American Geriatrics Society also recommends reducing opioid use in geriatric patients with trauma [11]. Thus, any method which can help to reduce the acute use of opioids for pain management will benefit the patient and the public at large. 

### 1.1. Importance

To deviate from the prevailing opioid-driven mindset, it is important to identify alternative analgesic agents. Regional nerve blockade for the early analgesic management of elderly patients with hip fracture has become important in the last decade [12]. As an emerging choice, femoral nerve block is considerably safe and can be rapidly administered under ultrasound guidance. In a systematic review, femoral nerve block showed no increased risk of cardiac complications, deep vein thrombosis, pulmonary embolism, myocardial infarction, surgical wound infection, nausea, or vomiting [6]. Therefore, ultrasound-guided femoral nerve blocks (USFNBs) may be part of a multimodal non-opioid analgesic regimen in elderly patients with hip fractures. 

Currently, evidence suggests that femoral nerve block yields superior outcomes to standard care; however, previous studies have used small samples or obtained temporally limited measures of meaningful pain relief yielded by femoral nerve block and opioids in elderly patients with hip fractures [13,14,15,16,17]. Moreover, there are limited studies on the use of lidocaine-based single-shot USFNB for patients with hip fractures [17]. 

### 1.2. Goals of This Investigation

To address these gaps in the literature, the current study aimed to investigate whether lidocaine-based single-shot USFNB reduces acute pain in elderly patients with hip fracture in the emergency department (ED). The primary outcome measure was the difference in opioid consumption between patients who received USFNB and those who were managed with conventional analgesics under the emergency physician’s discretion, a mode of treatment known as standard care.

## 2. Materials and Methods

### 2.1. Study Design and Protocol

This retrospective, single-center, observational study was conducted at an academic medical center with an ED. This ED manages approximately 41,000 annual visits, including approximately 20,000 visits by patients aged ≥60 years. The study protocol for the health record review was approved by the Institutional Review Board of Dalin Tzu Chi Hospital, Buddhist Tzu Chi Medical Foundation, Taiwan (No. 10903015). The study was structured according to recommendations detailed in the “Methods of Medical Record Review Studies in Emergency Medicine Research” [18].

### 2.2. Study Population and Setting

Elderly patients aged ≥60 years presenting in the ED with a diagnosis of traumatic hip fracture between 1 January 2017 and 31 December 2020, were included. We excluded patients who were aged <60 years, were non-communicative, were non-consenting, lacked follow-up within our health system or did not undergo hospitalization, had an allergy to opioids or local anesthetics, required management for ≥2 fractures within one visit, were diagnosed with avascular necrosis or hip infection, or had hemodynamic instability.

The variables and outcome were defined using the International Classification of Disease, 9th Revision, as well as codes, associated treatments, radiologist reports, and orthopedist consultation lists. Data were collected by an investigator from an electronic health record database using a standardized case record form. A second investigator collected data for 10% of a random selection of the total patient cohort. Both investigators were blind to the hypothesis. We calculated the ĸ-value for inter-observer reliability.

### 2.3. Interventions

USFNB was performed by emergency physicians (EPs) trained in emergency medicine, certified by the regional anesthesia training board. The EPs administered each USFNB when providers were available. The patients were divided into two groups by convenience non-probability sampling: the USFNB group and the standard care group. 

USFNB was performed using a hypodermic needle (NIPRO^®^ 21G × 70 mm) at the level of the inguinal area, lateral to the femoral artery, under real-time ultrasound guidance. Upon identification of the femoral nerve, the needle was inserted in-plane in a lateral to medial orientation, and the provider injected lidocaine once the needle tip was adjacent to the nerve (laterally, above, or below). We used 20 mL of 1% lidocaine for nerve block because this drug has a short onset time, which is adequate to relieve pain before surgical intervention [19]. Regimens of regional anesthesia with a longer duration of action are not preferred in our department because the delayed onset of regional anesthesia particularly impedes orthopedic assessment postoperatively. In the standard care group, patients received conventional standing and either parenteral or oral analgesics as needed (opioids, non-steroidal anti-inflammatory drugs, and acetaminophen) at the discretion of the treating EPs, according to the WHO analgesic ladder. The EPs were also instructed to aim for a 50% pain reduction or per patient request. They were encouraged to wait for at least 15 min after the study procedure before administering additional analgesia.

### 2.4. Measurements

Total opioid consumption was measured in morphine milligram equivalents (MME) administered over all ED visits during the study period. The treatment response was recorded according to the ED policy, i.e., pain intensity was assessed using the 11-point Numerical Rating Scale (NRS) at baseline; immediately after the analgesic treatment at 15, 30, 60, and 120 min after the dose was administered; every 2 h thereafter until discharge from the ED; and at the time of transfer from the ED and arrival at the orthopedic ward before surgery. They were also investigated through analysis of pain intensity difference (PID). PID represents the difference in NRS scores between any observation time and the baseline [14]. Time-to-event analysis was performed by considering a response to meaningful pain relief treatment as an event. Meaningful pain relief was defined as PID ≥ 4, which has been previously demonstrated to represent the clinically important measurement of pain outcomes, with a specificity of 93.6% [20]. For the USFNB group, the reference time point for treatment response was defined as the end of the procedure; for the standard care group, the start was at the time of medication administration. The time to meaningful pain relief was defined as the post-intervention time during which the patient started experiencing meaningful pain relief. 

### 2.5. Outcomes

The primary outcome measure was the difference in opioid consumption between the USFNB and standard care groups. The secondary outcomes included changes in the duration of ED stay, the duration of hospital stay, and the occurrence of complications. Complications were defined as vascular puncture (hematoma), permanent nerve injury, local anesthetic systemic toxicity, or any other adverse event. The obtained data were evaluated for multiple effectiveness aspects of USFNB compared with those of standard care.

### 2.6. Statistical Analysis

Patient characteristics and outcome measures are reported as means, standard deviations (SDs), 95% confidence intervals (CIs), and percentages, as appropriate. The Chi-squared test or Fisher’s exact test was performed to determine the categorical variables, and Student’s *t*-test was performed to determine the continuous variables. A *p*-value of <0.05 was considered statistically significant. Multivariable linear regression analysis was performed to analyze the outcome measures, including MME, duration of ED stay, and duration of hospital stay, with adjustments made for covariables including age, sex, body mass index (BMI), Charlson Comorbidity Index (CCI), initial pain score, chronic arthritis, chronic opioid use, and fracture type.

Since PID was measured as an unequally spaced time series, we used a continuous autoregressive time series model, AR (1), to model the dependencies between timepoints within each patient’s PID trajectory. An additional patient-specific random intercept was also added to reflect the variation in initial pain level between patients. The mean trajectory of PID was modeled with restricted cubic splines to allow for the non-linear relation between PID and time [21]. The difference in the trajectories between the USFNB group and the standard care group was evaluated using a likelihood ratio test with the significance level set at 0.05.

The time to onset of meaningful pain relief within 6 h after the dose initiation was determined using time-to-event analysis. A Kaplan–Meier plot was used to illustrate the time to meaningful pain relief, and a log-rank test was performed to compare the curves between the groups. A Cox proportional hazards regression model was used to compute the HRs and corresponding 95% Cis for achieving meaningful pain relief; multivariable Cox proportional hazards regression analysis was performed with adjustments made for covariables including age, sex, BMI, CCI, initial pain score, chronic arthritis, chronic opioid use, and fracture type. The time-to-event data were right censored when a patient’s event time (time to meaningful pain relief) was not observed, and the observation was imputed to have occurred later than the time when the last known value was recorded. For example, if a patient withdrew from the study 6 h after intervention without meaningful pain relief, then the event time was censored at 6 h. 

We excluded variables with >25% of the data missing or a cell count of ≤5. When the outcome data were missing, the appropriate statistical data imputation techniques were used to ensure the completion of the data set. All statistical analyses were conducted using STATA version 15.1 (StataCorp, College Station, TX, USA) and R Statistical Software (version 1.3; R Foundation for Statistical Computing, Vienna, Austria).

## 3. Results

### 3.1. Characteristics of the Study Participants

The data collected from 607 patients were included in the final analysis (USFNB group, 66; standard care group, 541) (Figure 1). In the standard care group, 305 patients were administered opioids (99% parenteral morphine) for acute pain management and 236 were administered non-opioid analgesics as needed.

The baseline characteristics are presented in Table 1. In the USFNB group, the average age of the patients was 80.30 years; 66.67% of female participants reported a higher initial mean pain score (*p* = 0.013). Overall, there was no significant difference between the two groups in age, sex, BMI, fracture type, and CCI.

### 3.2. Main Results

In the ED, patients in the USFNB group required 80% less MME than those in the standard care group (0.52 ± 1.47 mg vs. 2.57 ± 2.53 mg), after adjusting for age, sex, BMI, CCI score, initial pain score, chronic arthritis, chronic opioid use, and fracture type (*p* < 0.001; Table 2). There was no significant difference between the USFNB and standard care groups with respect to the duration of ED stay (*p* = 0.203) and hospital stay (*p* = 0.199; Table 2). No deaths, complications, or discontinuations due to adverse events were found to be related to the USFNB.

### 3.3. Comparison of Pain Scores between the USFNB and Standard Care Groups

To compare the analgesic efficacy of lidocaine-based single-shot USFNB with that of standard care in elderly patients with acute hip fracture, the data of 607 patients were analyzed using time-to-event analysis (time to meaningful pain relief) and pain trajectory. Meaningful pain relief was experienced by 71% and 67% of patients in the USFNB and standard care groups in 6 h, respectively. Time-to-event analysis of the outcomes showed that the median time to meaningful pain relief was significantly shorter in the USFNB group (96 min) than in the standard care group (185 min). Kaplan–Meier analysis showed that the difference in meaningful pain relief between the USFNB and standard care groups was statistically significant (log-rank test, *p* < 0.001; Figure 2). Patients in the USFNB group achieved meaningful pain relief 2.37-fold faster than those in the standard care group, as shown using univariate (HR = 2.37; 95% CI = 1.73–3.24, *p* < 0.001) and multivariable Cox proportional hazards regression models (adjusted HR = 2.37; 95% CI = 1.70–3.30, *p* < 0.001).

To evaluate individual PID values between the USFNB and standard care groups at every assessment point in the study, we used regression analysis to demonstrate the time course. We assessed non-linear relationships of time and PID by using restricted cubic spline with six knots. The pain-intensity reduction tended to be lower in the USFNB group than in the standard care group, but this was not statistically significant (*p* = 0.198; Figure 3). 

## 4. Discussion

In this retrospective, single-center cohort study, we observed that patients who underwent USFNB were able to significantly decrease opioid use in the ED (80% MME reduction in USFNB, *p* < 0.001). Patients who received USFNB for preoperative analgesia in the ED experienced 2.37-fold faster meaningful pain relief than those who received standard care (HR = 2.37; 95% CI = 1.73–3.24, *p* < 0.001). USFNB recipients also revealed a trend of lower reported pain scores than those who received standard care within 6 h, but this result was not statistically significant (*p* = 0.198; Figure 3). To the best of our knowledge, this is the first study to evaluate the effectiveness and efficacy of single-shot USFNB with lidocaine in an ED.

The American Geriatrics Society recommends reducing opioid use in geriatric patients with trauma [11]. However, in the ED, geriatric patients who require observation after injury are often prescribed opioids [22]. Studies have reported that high preoperative opioid exposure in the geriatric population increases mortality and the risk of readmissions after surgery [23,24]. In Taiwan, opioid misuse also remains an unsolved problem [25]. Educational intervention in trauma management could reduce the rate of opioid prescription and improve opioid epidemics [26]. As part of a multimodal analgesic intervention, the evidence suggests that ultrasound-guided regional anesthesia for acute pain management could reduce opioid use [14,27]. At present, professionals other than specialist anesthetists can safely administer regional anesthesia in the ED and prehospital settings [28]. In this retrospective study, we provided evidence supporting the notion that EP-performed ultrasound-guided regional anesthesia reduces opioid use (80% MME reduction with USFNB, *p* < 0.001) and does not prolong the duration of ED stay (*p* = 0.203) or hospital stay (*p* = 0.199).

In the ED, timely pain control is an important component of optimal pain management. Given its short onset time and short duration of action, regional anesthesia with short-acting regimens, such as lidocaine, is considered to help achieve good pain reduction [29]. A short-acting regimen is also desirable because it allows for better sensory examination after the end of surgery. However, few studies have compared the effectiveness of lidocaine-based femoral nerve block with that of standard care. Most of the current studies compared femoral nerve block with long-acting regimens to standard care [14,15,16] One study compared the effectiveness of lidocaine-based femoral nerve block with that of intravenous morphine sulfate for femoral neck fracture, and the pain score was not significantly different between the groups. The study involved performing a landmark femoral nerve block instead of the ultrasound-guided method [30]. Currently, ultrasound-guided regional anesthesia leads to a greater block quality than nerve stimulation or the landmark technique [31]. The results of our study provide evidence supporting the effectiveness of lidocaine-based single-shot USFNB as a non-opioid alternative for acute pain management in the ED (HR = 2.37; 95% CI = 1.73–3.24, *p* < 0.001). 

In this retrospective study, patients in the USFNB group reported a higher initial pain score than those in the standard care group (7.86 ± 0.99 vs. 7.36 ± 1.60, *p* = 0.013). The study by Olsen et al. demonstrated that patients with a higher initial pain score require greater pain reduction to reach a minimum clinically important difference than those with a lower initial pain score [32]. Based on the current concepts, the EPs in our study seemed to have stronger motivation to administer USFNB to patients with severe pain. This phenomenon was also compatible with the current evidence that USFNB provides more effective and faster pain control [14,16,33].

Pain measurement is most strongly associated with a patient-determined indicator of clinical importance. A cut-off point of PID ≥ 4 provided higher specificity to define meaningful pain relief [20]. Through time-to-event analysis (time to meaningful pain relief, defined as PID ≥ 4), the pain model in our study demonstrated assay specificity that enabled differential efficacy in patients who received USFNB and opioids. Overall, USFNB produced effective pain reduction in elderly patients with hip fracture (HR = 2.37; 95% CI = 1.73–3.24, *p* < 0.001). Furthermore, our results appeared robust with time-to-event analysis performed with a 3-point decrease (HR = 2.20; 95% CI = 1.60–3.01, *p* < 0.001) or a 5-point decrease (HR = 2.30; 95% CI = 1.66–3.18, *p* < 0.001). 

The mechanism of fast pain reduction with the use of peripheral nerve blocks is based on the hypothesis that peripheral nerve blocks inhibit the propagation of impulses in nerve terminals to inhibit the perception of pain by the cerebral cortex. Local anesthetics can temporarily block the transmission of pain. Theoretically, by providing a site-specific afferent neural block, peripheral nerve blocks should offer timely pain control compared with general anesthesia [34]. In peripheral nerve blocks, regimens such as lidocaine are different from longer-acting local anesthetics such as bupivacaine, levobupivacaine, and ropivacaine in the onset of action, duration of action, degree of motor blockade, and toxicity. Lidocaine 1% is enough to achieve successful femoral nerve block [19]. These beneficial effects could provide optimal conditions for timely and effective pain control. In fact, our study supports this theory. However, the effectiveness of pain control is moderate, and one third of patients in both groups did not achieve clinically significant changes in meaningful pain relief. We postulate that the variations in the innervation of the hip itself probably explain the uneven effect. In a recent study, a novel regional technique is reported: the pericapsular nerve group (PENG) block for relevant landmarks to target the hip articular branches of the femoral nerve and the accessory obturator nerve. The PENG block has been shown to provide better pain reduction and a potential motor-sparing effect than current regional anesthesia [35]. 

### Limitations

Despite these promising findings, there are several limitations. First, based on availability and willingness to participate, the non-probability convenience sampling method was used to select participants for this study, which caused a discrepancy between the patients who received USFNB and those who received standard care. The association between the intervention and the results may have been influenced by the challenges in identifying and controlling all the confounding factors, including the pain score recording at different time points in each group, dosage and route of opioid administration, and differences in treating EPs. This analysis allowed for control of only the most common measurable confounders; however, unknown factors, such as the increasing intention to decrease opioid use, may also have contributed to the decrease in MME, and this aspect is difficult to measure. Second, this study retains a potential risk of self-selection bias that should be considered when analyzing and interpreting the obtained data. Because this was a retrospective observational study, the patients included in the study could select their own treatment. Third, although all the treating EPs were instructed to manage pain according to the WHO analgesic ladder, our retrospective observational study is limited because of a potential risk in the use of variable analgesics caused by the EPs’ personal experience and preference, which can influence the analysis and interpretation of the data. Lastly, the study was completed at an urban level 1 trauma center, and the patients who visit this center may not be representative of all EDs. Since there are more opioid needs at higher level trauma centers, the results should be similar or better in other EDs [36].

## 5. Conclusions

This retrospective, single-center cohort study indicated that the use of lidocaine-based single-shot USFNB is a fast-acting, non-opioid analgesic alternative that provides prolonged analgesic effects. It can be used to effectively reduce pain while minimizing opioid requirements to control acute pain in geriatric patients with traumatic hip fracture in the ED. Further large, randomized studies are warranted to confirm our observations.

## Figures and Tables

**Figure 1 jcm-11-02778-f001:**
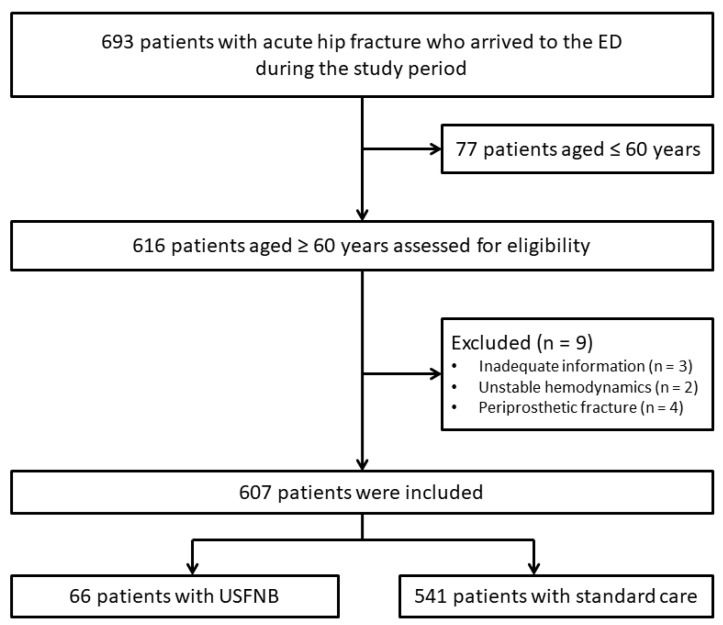
Flow diagram showing the selection of the study participants.

**Figure 2 jcm-11-02778-f002:**
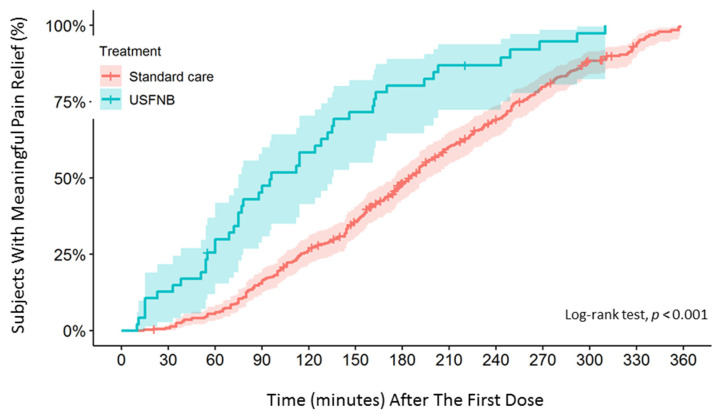
Kaplan–Meier plots for time to meaningful pain relief.

**Figure 3 jcm-11-02778-f003:**
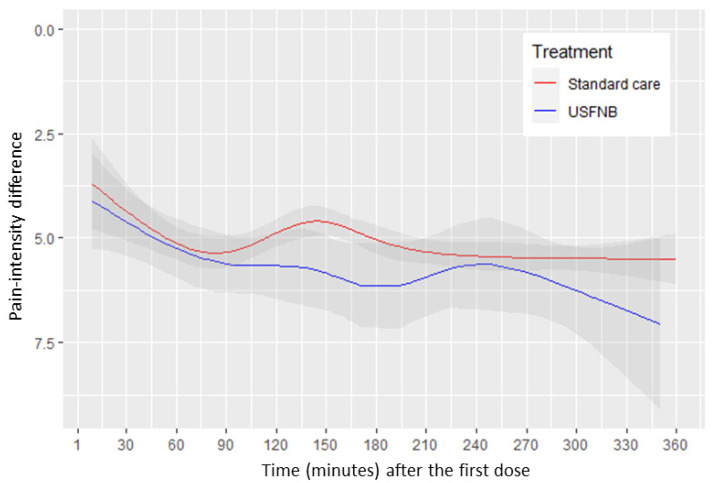
Pain score trajectory over time (minutes) between the USFNB group and standard care group.

**Table 1 jcm-11-02778-t001:** Characteristics of patients who did and did not receive USFNB.

Variables	USFNB (*n* = 66)	Standard Care (*n* = 541)	*p*-Value
Mean (SD) age (years)	80.30 (8.62)	79.69 (8.33)	0.579
Women, % (*n*)	66.67 (44)	65.99 (357)	0.913
Mean (SD) BMI	22.84 (4.49)	23.00 (4.20)	0.784
Fracture type			0.160
Extracapsular, % (*n*)	63.64 (42)	54.53 (295)	
Intracapsular, % (*n*)	36.36 (24)	45.47 (246)	
Mean (SD) of initial pain score	7.86 (0.99)	7.36 (1.60)	0.013
Triage *, %			0.845
Emergent	28.79	27.73	
Urgent	71.21	72.27	
Arrival (sent by EMT), %	46.97	38.45	0.181
Comorbidity			
DM, %	30.30	32.35	0.737
Hypertension, %	66.67	59.89	0.287
Cardiovascular disease, %	21.21	20.70	0.923
CKD, %	15.15	10.54	0.259
Liver disease, %	7.58	7.95	0.916
Surgical history, %	25.76	23.29	0.656
Chronic arthritis, %	13.64	6.10	0.023
Malignancy, %	13.64	8.69	0.190
Mean (SD) of CCIS	1.41 (1.26)	1.26 (1.33)	0.380
Chronic opioid users, %	6.06	5.91	0.962
Mortality, %	6.06	2.59	0.116

* Triage is known as Australasian Triage Scale, with 1 and 2 denoting emergent and 3–5 denoting urgent. Abbreviations: BMI, body mass index; CKD, chronic kidney disease; CCIS, Charlson Comorbidity Index Score; DM, diabetes mellitus; EMT, emergency medical technician; USFNB, ultrasound-guided femoral nerve block; SD, standard deviation.

**Table 2 jcm-11-02778-t002:** Outcomes of patients who did and did not receive USFNB.

		Morphine Milligram Equivalents (mg)			Duration of ED Stay (Hours)			Duration of Hospital Stay (Days)		
	No	Unadjusted Mean	Adjusted Difference	*p*-Value	Unadjusted Mean	Adjusted Difference	*p*-Value	Unadjusted Mean	Adjusted Difference	*p*-Value
USFNB	66	0.52 ± 1.47	−2.11 ± 0.32 ^†^	<0.001	7.01 ± 6.49	0.93 ± 0.73 ^†^	0.203	7.98 ± 3.91	−1.17 ± 0.91 ^†^	0.199
Standard care	541	2.57 ± 2.53	Reference		5.83 ± 5.43	Reference		9.28 ± 7.05	Reference	

Unless otherwise specified, the data are presented as mean ± SD. Adjustment for age, sex, body mass index, Charlson Comorbidity Index score, initial pain score, chronic arthritis, chronic opioid use; and fracture type. ^†^ The data are presented as effect size (SE). Abbreviations: ED, Emergency department; SD, standard deviation; SE, standard error; USFNB, ultrasound-guided femoral nerve block.

## Data Availability

The data are not publicly available due to regulations of the local institutional ethics board.

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
