# Peer review of "Ultrasound-Guided Femoral Nerve Block in Geriatric Patients with Hip Fracture in the Emergency Department"

_jcm, 2022, doi:10.3390/jcm11102778_

Round 1

Reviewer 1 Report

Review

This manuscript describes a retrospective observational study investigating the use of single shot ultrasound guided femoral nerve block (USFNB) as compared to conventional analgesia with regards to effectively managing pain and overall opiate consumption while in the emergency department. Statistical analysis revealed a significant decrease in opioid difference as measured by morphine milligram equivalents and decrease time to meaningful pain relief in the USFNB group. 

Abstract

  1. In the initial statement of the intend of this study in the abstract, recommend noting the nerve block is specifically a lidocaine injection, as mentioned in several other areas. Other regional anesthetics have different properties and are not being investigated.
  2. While the conclusion statements of more rapid and effective pain reduction are supported by significant differences in appropriate measures, there was no significant difference in pain scores at 6h between the 2 groups (P=0.198, Figure 3). Thus, the assertion that USFNB provides more sustained pain reduction is not supported by the evidence.

Introduction

  1. As this study is only investigating opiate use in the ED prior to surgery, background information concerning long term opioid abuse and opioid epidemic is not relevant. A narrower focus on the risks associated with acute opioid on the geriatric population provides more appropriate background information.

Materials and Methods

  1. In both the USFNB after the initial single shot and for the entirety of the duration in the standard care group, there is no standardized protocol for analgesia (NSAIDS vs opiates vs acetaminophen) and the management of analgesia was at the discretion of the emergency physician. Thus a level of variability in physician personal experience/preference is included. Also, the authors do not provide information on typical protocols nor information to document  how much variability there was.

Results – No Comments

Discussion

  1. Again, discussion of opioid epidemic and emergency department prescription of opioids appears to be irrelevant to the aim of analyzing pain relief in the emergency department prior to surgery for hip fractures.

Conclusion – No Comments

Author Response

Response to Comments from the Editor and Reviewers

Manuscript ID: jcm-1698295

We would like to thank the editor and the reviewers for their extensive assessment of our manuscript and for their helpful comments and suggestions. We have considered these remarks when revising our manuscript. In particular, we have made the following changes:

  1. We clarified the abstract.
  2. We clarified the statistical analysis in the Methods section.
  3. We reorganized the Discussion section. We have modified the Discussion section to focus on the preoperative opioid epidemic, and we have modified the limitations because of a potential risk in the use of variable analgesics caused by emergency physicians’ personal experience and preference.
  4. We have elaborated on several important issues in accordance with the editor’s and reviewers’ comments.

To adhere the formatting guidelines of Journal of Clinical Medicine, we have reduced the word count to <4000 words per the journal guidelines. Revisions addressing the editor’s and reviewers’ comments are highlighted in yellow. The revised manuscript has been edited by a native English speaker. The main corrections and point-by-point responses to the editor’s and reviewers’ comments are provided below. We are grateful for the opportunity to revise our manuscript. We hope that our responses and revisions have adequately addressed the reviewers’ concerns, and that the current version of the manuscript will now meet the high standards required for publication in our esteemed journal.

Sincerely,

Eric H Chou, MD

Department of Emergency Medicine,

Baylor Scott & White All Saints Medical Center, Fort Worth, TX, USA Address: 1400 8th Ave, Fort Worth, TX, 76104

Tel: 310.400.2306

Fax: 817.922.1954

Reviewer 2 Report

Dear Authors:

      I want to congratulate with You for Your work, and for the paper. The topic is interesting, the aim is well presented, the reasearch is well conducted, and results are clearly presented. Anyway, I would like to have some more informations. Here my points:

  • Table 1 shown characteristic of patients in both groups, and I can see a statistically significant difference in pain at admission and chronic arthritis as comorbidity. is there any correlation between this difference and results? how did You adjust results for these characteristics?
  • Can You comment between the use of Lidocaine 1% or Lidocaine 2% or a long acting local anestethic (Bupivacaine, for example)
  • Can You calculate / predict / say how many MME are 20 mL of Lidocaine 1%? did You consider it in the results?
  • My opinion is that the "opioid epidemics" is a problem in extra-hospital  treatment in chronic pain, not in peri-hospital (or intra-hospital) pain management for acute pain; do You suggest that the opioid epidemics is also an intra-hospital issue?

best regards, nm

Author Response

(The authors gave the same response as above.)
